# Point-of-Care Testing for Infectious Diseases Based on Class 2 CRISPR/Cas Technology

**DOI:** 10.3390/diagnostics13132255

**Published:** 2023-07-03

**Authors:** Shiu-Jau Chen, Chung-I Rai, Shao-Cheng Wang, Yuan-Chuan Chen

**Affiliations:** 1Department of Neurosurgery, Mackay Memorial Hospital, Taipei 10449, Taiwan; chenshiujau@gmail.com; 2Department of Medicine, Mackay Medical College, New Taipei City 25245, Taiwan; 3Health Care Business Group, Foxconn Technology Co., Ltd., New Taipei City 23680, Taiwan; barry.rai@gmail.com; 4Graduate Institute of Applied Science and Technology, National Taiwan University of Science and Technology, Taipei City 106335, Taiwan; 5Department of Psychiatric, Taoyuan General Hospital, Ministry of Health and Welfare, Taoyuan 33004, Taiwan; 6Department of Nurse-Midwifery and Women Health, National Taipei University of Nursing and Health Sciences, Taipei 112303, Taiwan; 7Department of Nursing, Jenteh Junior College of Medicine, Nursing and Management, Miaoli County 35664, Taiwan; 8Department of Medical Technology, Jenteh Junior College of Medicine, Nursing and Management, Miaoli County 35664, Taiwan; 9Program in Comparative Biochemistry, University of California, Berkeley, CA 94720, USA

**Keywords:** CRISPR, Cas9, Cas12a, Cas13, DETECTR, SHERLOCK, LOD

## Abstract

The early detection of infectious diseases and microorganisms is critical for effective disease treatment, control, and prevention. Currently, nucleic acid testing and antigen–antibody serum reaction are the two methods most commonly used for the detection of infectious diseases. The former is highly accurate, specific, and sensitive, but it is time-consuming, expensive, and has special technician and instrument requirements. The latter is rapid and economical, but it may not be accurate and sensitive enough. Therefore, it is necessary to develop a quick and on-site diagnostic test for point-of-care testing (POCT) to enable the clinical detection of infectious diseases that is accurate, sensitive, convenient, cheap, and portable. Here, CRISPR/Cas-based detection methods are detailed and discussed in depth. The powerful capacity of these methods will facilitate the development of diagnostic tools for POCT, though they still have some limitations. This review explores and highlights POCT based on the class 2 CRISPR/Cas assay, such as Cas12 and Cas13 proteins, for the detection of infectious diseases. We also provide an outlook on perspectives, multi-application scenarios, clinical applications, and limitations for POCT based on class 2 CRISPR/Cas technology.

## 1. Introduction

Infectious diseases are one of the major causes for global illness and death. Many people continue to suffer from infectious diseases, and some diseases are difficult to treat and prevent. To avoid microbial transmission in advance and treat infectious diseases promptly, it is crucial to accurately, sensitively, and rapidly detect pathogenic microorganisms. Various diagnostic methods and/or devices have been developed which can be classified in terms of their speed, portability, and readout. Currently, nucleic acid testing methods, including Southern blotting, Northern blotting, and real-time quantitative polymerase chain reaction (qPCR) and antigen (Ag)–antibody (Ab) serum reactions (e.g., complement fixation test, immunofluorescent assay, immunosorbent assay, etc.), are the two most common methods used for the diagnosis of infectious diseases and microorganisms. However, nucleic acid testing is time-consuming (more than 4 h or even 24 h), not portable, and troublesome (a lot of work, devices, and expertise are needed), while the results of Ag-Ab serum reactions are not reliable enough because their sensitivity and accuracy may be insufficient (Table 1).

A qPCR or quantitative reverse transcription polymerase chain reaction assay (qRT-PCR) can provide real-time detection of the product in each cycle in the PCR reaction. When the intensity of the fluorescent signal in a certain cycle reaches the present threshold value, the cycle number at this time is called the threshold cycle (Ct), and the Ct value is proportional to the initial DNA template [12,13,14,15]. The greater the amount of initial nucleic acid, the fewer the number of cycles that are needed to reach the threshold, that is, the smaller the Ct value. Taking the Ct value as the vertical axis and the initial template number as the horizontal axis to make a standard curve, the copy number of nucleic acid can be accurately calculated [12,13,14,15]. Therefore, qPCR or qRT-PCR is usually considered a more precise method for the detection of infectious diseases because it has the characteristics of a high accuracy, specificity, and sensitivity, as well as providing the Ct value and the limit of detection (LOD). However, it is time-consuming, expensive, and requires special devices/professional operators.

The most effective method to prevent the spread of diseases and guide proper treatment is to provide a quick and on-site diagnostic test that is accurate, sensitive, convenient, cheap, and portable, such as point-of-care testing (POCT). POCT can provide fast and feasible diagnostic results near patients and thereby acts as a personal exploratory detector for infectious diseases [16,17,18]. Consequently, it is needed to develop an effective and efficient POCT to diagnose infectious diseases more sensitively and accurately than Ag-Ab serum reactions, and more rapidly and conveniently than qPCR (or qRT-PCR). Fortunately, clustered regularly interspaced short palindromic repeat (CRISPR)-based technology is able to provide a strategy to detect infectious diseases and microorganisms rapidly, conveniently, robustly, and sensitively. For example, CRISPR/Cas12 and CRISPR/Cas13 can be rapid, sensitive, specific, portable, and economical methods for the diagnosis of bacteria and viruses. These CRISPR-based diagnostics are dependent on type V CRISPR/Cas12, otherwise called DNA Endonuclease-Targeted CRISPR Trans Reporter (DETECTR) or type VI CRISPR/Cas13, otherwise called Specific High-sensitive Enzymatic Reporter UnLOCKing (SHERLOCK) [19,20,21]. Both of these Cas12 and Cas13 enzymes exhibit nonspecific endonuclease activity in trans after binding to a specific cis target via programmable CRISPR RNAs (crRNAs). In this article, we provide a review of class 2 CRISPR/Cas-based technologies for POCT, as well as their perspectives, limitations, and possible strategies to improve them.

## 2. CRISPR/Cas System

### 2.1. CRISPR Brief

CRISPR, originally discovered in the genome of *Escherichia coli*, is a regular sequence containing many small repeating DNA segments with equal intervals between these repeating segments [22,23]. It is a DNA sequence consisting of multiple, short, and direct repeats, each repeat containing about 30 base pairs called spacer DNA. CRISPR is a specific DNA sequence found in both eubacteria and archaea whose genomes contain DNA fragments from viruses that infected the bacteria previously [22,23]. CRISPR is an adaptive immunity system produced in the host bacteria to resist foreign genetic materials such as plasmid or bacteriophage DNA [24,25]. Repetitive DNA sequences are found in bacteria with “spacer” DNA sequences in between the repeats that exactly match viral sequences. Once the same DNA sequence enters the bacteria again, bacteria can subsequently transcribe these DNA elements to RNA upon viral infection. The bacteria have a strong memory against the invaded DNA and generate acquired immune responses to break down foreign DNA via complementary crRNA [24,25].

### 2.2. CRISPR Associated (Cas) Proteins

Cas proteins can be classified into two classes (1 and 2) and six subtypes (I~VI) by the molecular mechanisms. Class 1 Cas proteins (type I, type III, and type IV) use multiple different proteins together with crRNA to build a functional endonuclease, whereas class 2 Cas proteins (types II Cas9, type V Cas12, and type VI Cas13) only use a single protein, which is a nuclease effector guided by crRNA to target nucleic acids [26]. 

The structure of Cas9 protein (formerly called Cas5, Csn1, or Csx12) is bilobed, consisting of the target recognition domain and nuclease lobe which contains the endonucleases RvuC and HNH and a carboxy-terminal region that recognizes a region adjacent to the protospacer adjacent motif (PAM). Cas9 is a DNA endonuclease which is guided by two RNAs-CrRNA and trans-activating crRNA (tracrRNA). Cas9 is able to mediate RNA-guided DNA targeting and cleavage. The mature crRNA base-pairs to the tracrRNA to form a two-RNA structure. The crRNA/tracrRNA hybrid acts as a single-guiding (sg) RNA to direct the Cas9 to make double-stranded breaks (DSB) in target DNA [27,28]. After the Cas9 protein binds with the sg RNA, it will guide the endonuclease activity to the region adjacent to the PAM. The Cas9-sgRNA complex creates a blunt end with DSB upstream 3 nucleotides from PAM at the cleavage site [27,28]. The change from a tripartite to a bipartite system is crucial to make Cas9 practical to use for genome editing and also makes its potential use as a point-of-care diagnostic feasible [29] (Table 2).

Cas12 protein, including isoform Cas12a (Cpf1) and Cas12b (C2c1), is a smaller and simpler DNA endonuclease than the Cas9 protein. It uses a single RuvC catalytic domain for guide RNA-directed double-stranded (ds) DNA cleavage. Unlike Cas9, Cas12 enzymes recognize a T nucleotide-rich region to catalyze the maturation of their own guide crRNA and generate a PAM distal DSB with staggered 5′ and 3′ ends [30,31,32]. Therefore, Cas12a does not require a tracrRNA and creates a sticky end to form DNA DSB downstream 19 to 23 nucleotides from PAM rather than a blunt end (upstream 3 nucleotides like Cas9) at the cleavage site. Cas12 can be searched in the genome database of bacteria, and its gene appears in the genomes of many species [33]. Cas12 is a single RNA-guided endonuclease lacking tracrRNA and utilizes a T-rich PAM. (Table 2)

Cas13, a novel type of RNA-targeting enzyme including Cas13a (formerly called C2c2), Cas13b, Cas13c, and Cas13d subtypes, was identified in the computational analysis. It is the only Cas protein known to exclusively bind and cut foreign RNA [34,35,36]. Originally, Cas13a was found to programmatically bind and cleave RNA to manipulate RNA [37]. The cleavage is mediated by catalytic residues in the two conserved higher eukaryotes and prokaryotes nucleotide (HEPN)-binding domains [37]. Further computational and biochemical studies of Cas13 have resulted in better understanding all subtypes. Cas13 can be reprogrammed to cleave a target single-stranded (ss) RNA through a short guide RNA complementary to the target sequence. Similar to Cas9, Cas13 complexes with the sgRNA contain about 64 nucleotides via the recognition of a short hairpin in the crRNA to code for target specificity. The target specificity is encoded by a 28~30 nucleotide spacer that is complementary to the target region [34,35,36,37]. Additionally, Cas13 can recognize and cut a target transcript, leading the nearby transcript to degrade nonspecifically regardless of complementarity to the spacer [36,37] (Table 2).

**Table 2 diagnostics-13-02255-t002:** Comparison of Class 2 CRISPR associated (Cas) proteins.

Cas protein	Cas9	Cas12	Cas13
Type	II	V	VI
Source of microbes	*Streptococcus pyogenes, Streptococcus thermophilus, Staphylococcus aureus, Neisseria meningitidis, Campylobacter jejuni*	*Francisella novicida, Acidaminococcus sp., Lachnospiraceae sp., Prevotella sp.*	*Leptotrichia buccalis, Leptotrichia shahii, Ruminococcus flavefaciens, Bergeyella zoohelcum, Prevotella buccae, Listeria seeligeri* *, Porphyromonas gulae*
Cleavage	A blunt end with DNA DSB upstream 3 nucleotides from PAM	A sticky end with DNA DSB downstream 19 to 23 nucleotides from PAM	Single-stranded (ss) RNA
Size	1000~1600 amino acids	1100~1300 amino acids	900~1300 amino acids
Guide spacer length	18–24 nucleotides	18–25 nucleotides	12–30 nucleotides
Total guide length	~100 nucleotides (sgRNA)	42–44 nucleotides	52–66 nucleotides
PAM sequence	3-NGG (SpCas9, N is any nucleotide); 3-NNGRRT (SaCas9, R is A or G); 3- NNNNGATT (NmCas9); 3-NNNVRYAC (CjCas9, V is A, G, or C; Y is T or C)	5-TTTN (FnCas12a)	3-H (LshCas13a); 5-D and 3-NAN or NNA (BzCas13b); none (RfCas13d)
RNA needed	crRNA+ tracrRNA (single-guide RNA)	crRNA	No
Application	Gene editing, diagnostics	Gene editing, diagnostics	Transcript knockdown, transcript imaging, RNA editing, diagnostics
Reference	[29,38,39,40,41]	[42,43,44]	[45,46,47]

Abbreviation: Double-stranded break (DSB); protospacer adjacent motif (PAM).

### 2.3. Application of the Class 2 CRISPR/Cas System

The CRISPR/Cas9 system can be used as an RNA-programmable DNA targeting and gene editing platform such as silencing, enhancing, or modifying specific genes. It can be simplified by a synthetic sgRNA which resembles the natural dual tracrRNA-crRNA structure. The presence of PAM is strictly required for target recognition, and the subsequent formation and strand cleavage are induced by complementary base pairing between the sgRNA and target DNA, Cas9-DNA interactions, and related conformational changes. The CRISPR/Cas9 system has been extensively worked as a genetic engineering or editing tool in a wide range of organisms [48,49,50,51,52].

The CRISPR/Cas12 system has also been applied for gene editing [53,54]. Additionally, it has been successfully used to detect a variety of microorganisms [55,56]. CRISPR/Cas12a-based DETECTR enabled the rapid and specific detection of patient specimens, thereby providing a simple platform for the molecular diagnosis of infectious diseases. Recent advances in CRISPR/Cas12-based detection have developed strategies for the improvement of sensitivity, optimization of integrated detection, simplification of detection mode, and quantitative detection [57].

The CRISPR/Cas13 system can be engineered for RNA knockdown and binding to become a flexible platform for studying RNA in mammalian cells. It is referred to as RNA editing for Programmable A to I Replacement (REPAIR) without strict sequence restriction and can be used to edit full-length transcripts [34,58]. It can be reprogrammed and applied to against pathogenic RNA viruses in eukaryotes or regulate the gene expression, promoting the knockdown of mRNAs, circular RNAs, and noncoding RNAs [37]. Additionally, the CRISPR/Cas13 system provides a novel tool to rapidly detect pathogenic microorganisms in combination with nucleic acid extraction, isothermal amplification, and product detection [59].

## 3. Diagnosis of Infectious Diseases Based on Class 2 CRISPR/Cas Systems

The development of nonspecific trans-cleavage activities of several Cas proteins (e.g., Cas12 and Cas13) has successfully facilitated many CRISPR/Cas-based molecular diagnostic technologies for microbial infection, including several viruses and bacteria, showing advantages in rapidity, specificity, sensitivity, and convenience. 

### 3.1. Acute Respiratory Syndrome Coronavirus 2 (SARS-CoV-2)

SARS-CoV-2, the pathogen of Coronavirus Infectious Disease 2019 (COVID-19), is a coronavirus with a positive sense single-stranded RNA core and helical symmetry of the coat protein (capsid) and the envelope. Its genome codes for four structural proteins, including the nucleocapsid (N) protein, membrane (M) protein, spike (S) protein, and envelope (E) protein, as well as some non-structural proteins [60,61]. qRT-PCR and the antigen (e.g., N or E protein) testing of SARS-CoV-2 are currently major methods for COVID-19 detection [62]. qRT-PCR is considered to be more sensitive than antigen testing, but it is time-consuming, expensive, and troublesome. Antigen testing is rapid, cheap, portable, and simple and can be used for POCT, but its sensitivity is considered to be less than qRT-PCR.

In July 2020, Broughton et al. demonstrated a rapid (less than 40 min), easy, and accurate CRISPR/Cas12-based lateral flow assay to detect the E and N genes of SARS-CoV-2 from respiratory swab RNA extracts [63]. This assay implements simultaneous reverse transcription (RT) and isothermal amplification using loop-mediated amplification (LAMP) for RNA extract. They validated this method using contrived reference samples and clinical samples from patients, including 36 patients with COVID-19 and 42 patients with other viral respiratory infections [63]. The results revealed that the CRISPR/Cas12a-based DETECTR assay can provide a visual and faster alternative method with 95% positive predictive agreement and 100% negative predictive agreement compared with the qRT-PCR assay [63].

In December 2020, Xiong et al. developed a CRISPR/Cas12a system to detect the open reading frame (ORF) 1ab and N genes of SARS-CoV-2, for which the results can be observed by the naked eye or evaluated using a fluorescent reader [64]. It is a rapid sample processing approach combining with recombinase polymerase amplification (RPA). They evaluated this system using 22 clinical samples originally diagnosed by the RT-qPCR assay [64]. The results showed 100% consistency between their assay and the RT-qPCR readouts for both negative and positive samples. Additionally, the results can be achieved within 50 min with a high sensitivity (1~10 copies per reaction) [64]. The application of CRISPR/Cas12a-based diagnostic method may help to control COVID-19 pandemic effectively.

In January 2021, Fozouni et al. reported the development of an amplification-free CRISPR/Cas13a-based assay to detect the N gene of SARS-CoV-2 from nasal swab RNA of patients within 30 min [65]. This assay also accurately detected pre-extracted RNA from a set of positive clinical samples within 5 min. The sensitivity of this assay can be achieved in ∼100 copies/μL and read with a mobile phone microscope, allowing for a portable and sensitive readout [65]. They used crRNAs to target SARS-CoV-2 RNAs for the improvement of sensitivity and specificity, and the direct viral load was quantified using enzyme kinetics [65]. This assay potentially provides a rapid, accurate, low-cost, point-of-care method to screen for SARS-CoV-2 by combining the reader device based on a mobile phone.

In February 2021, Sun et al. developed a one-tube detection platform using reverse transcription, recombinase polymerase isothermal amplification (RT-RPA), and CRISPR/Cas12a-based DETECTR technologies (OR-DETECTR) to detect SARS-CoV-2 [66]. They designed RT-RPA primers of the RNA-dependent RNA polymerase (RdRp) gene and N gene and then optimized reaction components to let the detection process be carried out in one tube and completed within approximately 50 min. SARS-CoV-2 and influenza A (H1N1) can be detected with a low LOD of 2.5 copies/µL input using the RNA standard and 1 copy/µL input using pseudoviruses [66]. The results showed that OR-DETECTR is 100% consistent with real-time RT-PCR after analyzing six samples from SARS-CoV-2 patients, eight samples from patients with fever but without SARS-CoV-2 infections, and one mixed sample from 40 negative controls [66]. The lateral flow assay based on OR-DETECTR can be used to detect COVID-19 with an LOD of 2.5 copies/µL input and might be a platform with a short detection time, reduced equipment requirements, and without aerosol contamination [66].

Therefore, the class 2 CRISPR/Cas-based assay has the potential to be POCT for the detection of SARS-CoV-2 to serve as an alternative strategy for the currently used traditional methods.

### 3.2. Influenza Virus

The influenza virus that infects humans has three types: A, B, and C. The common types, A and B, are spherical in appearance, and type C is filamentous and relatively rare (usually only cause mild symptoms in children). The basic structure of influenza viruses is similar. Their genomes are negative sense, single-stranded RNA. Their outermost structure is the envelope, and the surface mainly contains two glycoproteins: hemagglutinin (HA, sialidase) and neuraminidase (NA) [67]. The HA binds to sialic acid on the surface of the host’s oral and nasal respiratory epithelial cells. After the HA is cleaved by protease, the RNA genome enters the host cell through endocytosis [67]. After the virion matures, NA cleaves the sugar on its surface to assist viruses release from the host cell to infect new host cells [67]. The matrix protein 2 (M2) is a proton-selective protein on the envelope (only in type A) and it forms an ion channel, allowing the viral RNA to be released and enter host cells [67]. The current diagnostic methods of influenza viruses include virus culture, antigen testing, antibody testing, immunofluorescence testing, and qRT-PCR [68]. Virus culture, antibody testing, immunofluorescence testing, and qRT-PCR are time-consuming and troublesome. Antigen testing is specific, rapid, and convenient, but its sensitivity may be not enough.

In March 2021, Park et al. developed a CRISPR/Cas12a-system-based DETECTR to detect the M gene of influenza A virus (IAV) and the HA gene of influenza B virus (IBV) within 75 to 85 min by coupling RT-RPA and RT-LAMP [69]. The limit of detection of viral titers is one plaque forming unit (PFU) per reaction without exhibiting cross-reactivity. The results were verified using a lateral flow strip assay and no additional analytic devices are needed [69]. They established RT-RPA- and RT-LAMP-coupled DETECTR-based diagnostic tests for the detection of IAV and IBV using fluorescence and lateral flow assays rapidly, specifically, and sensitively [69]. This diagnostic test using DETECTR can be used to distinguish IAV and IBV infections, with the capacity to prevent and control the transmission of influenza epidemics and pandemics.

In October 2022, Zhou et al. reported a colorimetric biosensor for the influenza H1N1 virus assay based on the CRISPR/Cas13a system and hybridization chain reaction (HCR) [70]. They established and verified CRISPR/Cas13a-based visual influenza H1N1 viruses using the label-free and isothermal detection method. The trans-cleavage activity of Cas13a would be activated by targeting the RNA of influenza H1N1 virus to initiate HCR to copiously generate G-rich DNA [70]. A colorimetric reaction would be catalyzed by abundant G-quadruplex/hemin formed in the presence of hemin. The colorimetric biosensor showed a linear relationship over the range from 10 pM to 100 nM. The method has excellent specificity and sensitivity, with an LOD of 0.152 pM [70]. This novel method could be suitable for basic research and the clinical detection of influenza virus for POCT because it is fast, efficient, and does not require a variable-temperature environment and special expensive equipment.

### 3.3. SARS-CoV-2 and Influenza Virus

In February 2021, Mayuramart et al. applied the CRISPR/Cas12a system to detect the S gene of SARS-CoV-2, the matrix (M) gene of IAV, and the M gene of (IBV) [71]. This is a practical application for diagnosing patients with COVID-19, influenza A, and influenza B with a limit of detection 10, 10^3^, and 10^3^ copies per reaction, respectively [71]. The results suggested that the detection of SARS-CoV-2 using the RT-RPA with CRISPR/Cas12a assay is 96.23% in sensitivity and 100% in specificity. The sensitivity for IAV and IBV detections was 85.07% and 94.87%, respectively. Moreover, the specificity for IAV and IBV detections was both about 96% [71].

Therefore, CRISPR/Cas12a-based technology is a platform to serve as a POCT which is fast, portable, simple, and disposable for the detection of SARS-CoV-2 and influenza viruses.

### 3.4. Human Papillomavirus (HPV)

HPV, a double-stranded DNA virus, usually causes no symptoms, but it may persist and results in either precancerous lesions or warts in some cases. Nearly all cervical cancer is due to two strains, HPV-16 and HPV-18, accounting for about 70% cases. HPV-6 and HPV-11 are common causes of genital warts and laryngeal papillomatosis. The current diagnostic methods of HPV include Pap smear (Papanicolaou-stain), DNA hybridization, and mRNA detection by chemiluminescent analysis or flow cytometry and qPCR [72,73,74]. DNA hybridization, mRNA detection, and qPCR are time-consuming, troublesome, and not portable. Pap smear is an invasive and risky detection method.

In June 2021, Gong et al. combined the isothermal RPA method with CRISPR/Cas12-based technology to detect 13 types of especially high-risk types of human papillomavirus (HR-HPV) in a single reaction [75]. They designed 13 pairs of RPA primers to target the L1 conservative region of HR-HPVs. The results were obtained within 35 min with sensitivity (500 copies per reaction) [75]. Additionally, the developed assay showed 100% positive and negative agreements compared with the qPCR assay [75]. This RPA-CRISPR/Cas12a assay represents great advances and potential to address the key challenges facing HPV diagnostics.

In December 2022, Zheng et al. established a CRISPR-Cas12a/Cas13a dual-channel system combined with multiplex recombinase-aided amplification (RAA) to rapidly detect HPV16/18 for the screening of cervical cancer [76]. They designed a portable fluorescence imaging assay that can distinguish the test results directly by the naked eye or through cell phone imaging. The results showed that the LOD for HPV16 and HPV18 was both 10° copies/μL [76]. After this dual-channel assay was validated with 55 clinical samples, it revealed 97.06% sensitivity, 100% specificity, 100% positive predictive value, and 96.55% negative predictive value [76]. The fluorescence imaging assay is comparable to those of the real-time fluorescent RAA-based CRISPR-Cas12a/Cas13a dual-channel assay [76].

Therefore, the class 2 CRISPR/Cas (Cas12 and Cas13)-based assay has the potential to be POCT for HPV detection, which is fast, portable, specific, sensitive, and accurate.

### 3.5. Hepatitis B virus (HBV)

HBV, a double-stranded DNA virus, is mainly transmitted by exposure to infectious blood or body fluid and causes a type of viral hepatitis. It can result in both acute and chronic infections and even life-threating liver cirrhosis and cancer. The current diagnostic methods of HBV include serum or blood tests that detect viral antigens (e.g., HBsAg, HBcAg, HBeAg), anti-HBV antibodies produced by the host, and qPCR for viral load [77,78]. Interpretation of the antigen and antibody assay is complex, time-consuming, and not portable. qPCR is time-consuming and troublesome.

In May 2021, Ding et al. developed a POCT assay for HBV based on LAMP-Cas12a and solved the problem of point-of-care testing within 10 min, particularly for sample nucleic acid extraction [79]. They used lateral flow test strip technology based on LAMP-Cas12a to achieve results visible by the naked eye and achieved real-time high-sensitivity detection via fluorescence readout. The fluorescent-readout-based Cas12a assay detected HBV with an LOD of 1 copy/μL within 13 min, while the lateral flow test strip only takes 20 min [79]. After validation evaluation of 73 clinical samples, the sensitivity and specificity of the LAMP-Cas12a-based assay, including the fluorescence readout and lateral flow test strip method, reached 100% [79]. The LAMP-Cas12a-based HBV assay with minimal equipment requirements provided rapid and accurate test results which were fully comparable to qPCR [79]. Additionally, the method was highly specific and resistant to interference after continuous optimization [79]. This study showed that an HBV POCT based on LAMP-Cas12a has significant values in the prevention and control of hepatitis B.

In April 2022, Zhang et al. amplified specific covalently closed circular DNA (cccDNA) responsible for persistent HBV infection by rolling circle amplification (RCA) and PCR to detect the target gene using the CRISPR/Cas13a-based assay [80]. They established a novel CRISPR/Cas13a-based assay which was further clinically validated to detect cccDNA. After the amplification of RCA and PCR, 1 copy/μL HBV cccDNA was detected by CRISPR/Cas13-assisted fluorescence readout [80]. They detected 20, 4, 18, 14, and 29 positive samples in liver tissue samples from 40 hepatitis B patients using five methods, respectively: droplet digital PCR (ddPCR), qPCR, RCA-qPCR, PCR-CRISPR, and RCA-PCR-CRISPR [80]. However, HBV cccDNA was completely undetected in the 20 blood samples of hepatitis B patients using the same methods as above [80]. The results demonstrated that the CRISPR/Cas13a-based assay was highly sensitive and specific for the detection of HBV cccDNA, showing a promising alternative for the accurate detection and antiviral therapy treatment of HBV infections [80].

Therefore, the class 2 CRISPR/Cas (Cas12 and Cas13)-based assay has the potential to be POCT for HBV detection, which is fast, portable, specific, sensitive, and accurate, serving as an alternative method for current assays.

### 3.6. Hepatitis C virus (HCV)

HCV, a single-stranded RNA virus, is mainly transmitted by contacting infectious blood or body fluid. The virus can cause both acute and chronic hepatitis, ranging from a mild illness to a serious, lifelong, and threating illness including liver cirrhosis and cancer. The current diagnostic methods of HCV include serum or blood tests that detect anti-HCV antibodies produced by the host and qRT-PCR for viral loads [81,82,83]. Interpretation of the antibody assay is complex, time-consuming, and not portable. qRT-PCR is time-consuming, troublesome, and not portable.

In June 2022, Kham-Kjing et al. developed and validated a fast and accurate assay for the rapid detection of HCV RNA based on an RT-LAMP with the CRISPR/Cas12a assay for the recognition of specific HCV RNA sequences [84]. The amplified products after the cleavage reactions could be seen by the naked eye on lateral flow strips or evaluated with a fluorescence detector. They tested clinical samples from patients infected with HCV, HIV, or HBV or from healthy people [84]. For the sensitive test, of 100 plasma samples with known HCV viral loads, 93 samples were positive, as detected by RT-LAMP-coupled CRISPR/Cas12 assay with both readouts (lateral-flow-based and fluorescence-based readouts) [84]. Following retesting, only four samples remained negative with HCV viral loads of 3.96, 4.06, 4.85, and 6.58 Log_10_ IU/mL, respectively [84]. This means the sensitivity of the first round and the second round is, respectively, 93% and 96% in both readouts of the RT-LAMP-coupled CRISPR–Cas12 assay. For specificity testing, 30 non-HCV templates were used, including 10 HBV-, 10 HIV-infected individuals, and 10 healthy blood donors, respectively [84]. All samples showed no “Test” band on the lateral flow strip and no signal of fluorescence. The results suggested that the sensitivity and specificity of the RT-LAMP combined with the CRISPR/Cas12 assay was 96% and 100%, respectively, and showed 97% agreement, compared to the reference method [84]. This assay detected HCV RNA with an LOD of 10 ng/µL (an estimated 2.38 Log_10_ IU/mL) and may represent an affordable and reliable POCT for the identification of active HCV patients in low-resource settings [84].

Therefore, the CRISPR/Cas12-based assay is able to be POCT for HCV detection, which is fast, portable, specific, sensitive, and accurate, serving as an alternative method for current assays.

### 3.7. Staphylococcusaureus

*Staphylococcus aureus* is a Gram-positive spherically shaped bacterium. It is a usual member of the microbiota in the body and frequently found in the upper respiratory tract and on the skin. However, it sometimes causes skin infections, food poison, bone and joint infections, and even bacteremia. The current diagnostic methods of *Staphylococcus aureus* include bacteria culture of specimens for patients and qPCR [85,86]. These two methods are time-consuming, troublesome, and not portable.

In March 2022, Li et al. developed a platform for the diagnosis of methicillin-resistant *Staphylococcus aureus* (MRSA) by integrating RPA with the Cas12 system into one tube [87]. They used the one-tube RPA-CRISPR/Cas12a platform to achieve visual MRSA detection within 20 min. Based on this assay, the results were visualized by lateral flow test strips and fluorescent-based methods, including real-time and end-point fluorescence [87]. The sensitivity of lateral flow test strips ranged from 10 to 100 copies, and the fluorescence method was 10 copies [87]. After analyzing 23 samples from clinical MRSA isolates, the results revealed that the coincidence rate of the fluorescence method and lateral flow test strips was 100% and 95.7%, respectively, compared with qPCR [87].

Therefore, the one-tube RPA-CRISPR/Cas12a platform is an effective, rapid, accurate, and contamination-free method for MRSA diagnosis, showing potential for practical POCT applications of *Staphylococcus aureus*.

### 3.8. Mycobacterium Tuberculosis

*Mycobacterium tuberculosis* is a species of pathogenic bacteria and causes tuberculosis. Its physiology is highly aerobic and primarily a pathogen of the mammalian respiratory system, infecting the lungs. The current diagnostic methods include chest X-ray, tuberculin skin test, acid-fast stain, bacteria culture of specimens from patients, and qPCR [88,89]. These methods are all time-consuming, troublesome, and not portable.

In September 2021, Wang et al. designed the LAMP amplicons containing a specific PAM site for CRISPR/Cas12a recognition to activate its corresponding effector upon the CRISPR-Cas12a/gRNA/target DNA complex produced [90]. The single-stranded DNA (ssDNA) reporter molecules were rapidly degraded due to the trans-enzyme cleavage of CRISPR/Cas12a. The ssDNA could then be seen by the naked eye on a lateral flow biosensor or measured using a real-time fluorescence device [90]. Loop-mediated isothermal amplification coupled with the CRISPR/Cas12a-mediated diagnostic assay (LACD) made any target sequence detectable without the existence of PAM sites. This method was validated on the *Mycobacterium tuberculosis* complex [90]. This assay could also be applied to detect a variety of target sequences on other microorganisms as long as redesignation of the engineered LAMP primers occurs [90].

Therefore, the CRISPR/Cas12a-based assay has the potential to be POCT for *Mycobacterium tuberculosis* detection which is fast, portable, specific, sensitive, and accurate, serving as an alternative method for current assays.

## 4. Perspectives and Limitation of CRISPR-based Diagnosis for Point-of-Care Testing

The CRISPR/Cas system provides a portable diagnostic method which combines the advantages of qPCR/qRT-PCR and Ag-Ab serum reactions in terms of rapidity, accuracy, specificity, and convenience. Some class 2 CRISPR/Cas-based assays have been strictly validated against many targets, showing promise in detecting various microorganisms, including viruses, bacteria, parasites, chlamydia, and fungi. Currently, certain technologies are not only under basic research or clinical trial, but some of them have been advanced to clinical application for POCT. Class 2 CRISPR/Cas-based nucleic acid detection has been gradually commercialized for product marketing and shown to be promising for POCT in clinical application. For example, the CRISPR/Cas12 diagnostic assays developed by Mammoth Biosciences [63] have been granted approval for emergency use authority (EUA) by the US Food and Drug Administration (FDA) for the clinical detection of SARS-CoV-2 in Jan 2022 [91]. However, CRISPR/Cas-based assays still have some limitations, described as follows.

First of all, the readout of the CRISPR/Cas-based assay is visible by the naked eye like the Ag-Ab serum reaction, but it is only an estimative, not a precise result. The diagnostic rapidity of the CRISPR/Cas-based assay is the same as the Ag-Ab serum reaction in most cases, but its accuracy, specificity, and sensitivity is generally better than the Ag-Ab serum reaction for detecting microorganisms. The diagnostic specificity and accuracy of the class 2 CRISPR/Cas-based assay are almost the same as qPCR/qRT-PCR and ddPCR; however, it is only semiquantitative and cannot provide a Ct (threshold cycle) value of detection. Additionally, the class 2 CRISPR/Cas-based assay is less sensitive (LOD is higher) than qPCR and ddPCR, and the data obtained from this assay are not linear. If we would like to obtain more precise results, performing qPCR/qRT-PCR or ddPCR is required to double check and confirm the results. Though qPCR/qRT-PCR and ddPCR are able to provide a relative and/or absolute quantitation for nucleic acids detection, their application is limited by long-term detection and the requirement of heavy thermal cycling equipment. Therefore, it is needed to continue to lower the LOD and explore the data linearity of class 2 CRISPR/Cas-based assays using Cas12 and Cas13 proteins. Fortunately, a novel CRISPR/Cas13-based assay without amplification detection of SARS-CoV-2 has been established and can replace fluorescence microscopy with mobile phone-based fluorescence for quantitative detection [65].

Second, not all target genes (specific DNA/RNA sequences) for the detection of infectious diseases can be easily selected, and the selection process is quite complex. To maintain the detection stability long-term, it is necessary to choose target genes that are not easy to mutate. For example, the N gene and/or E gene of SARS-CoV-2 are usually chosen as the target genes for COVID-19 detection in the class 2 CRISPR/Cas-based assay because they are more stable than the S gene. However, the option of suitable target genes is troublesome, difficult, and even unavailable sometimes. Moreover, the target gene for detection is only relatively, not always, stable compared with other genes considered easy to change. To ensure the detection accuracy, precision, and stability of the class 2 CRISPR/Cas-based assay, the stability of target genes may need to be checked in a certain microorganism, causing a specific infectious disease periodically.

Third, the precise designation of highly specific and efficient crRNAs and isothermal primers is critically needed for the practical utilization of class 2 CRISPR/Cas-based technologies. This design process requires intensive manual curation and stringent parameters to minimize off-target detection and also preserve the detection of other strains. Therefore, it is essential to have a single and streamlined bioinformatics platform for rapidly designing crRNAs for the CRISPR/Cas platform. Fortunately, PrimedSherlock, an automated and computer-guided process for optioning highly specific crRNAs and primers for targets of interest, has been developed [92]. As a freely accessible software package, PrimedSherlock could significantly enhance the efficiency of the CRISPR/Cas12-based diagnostic assay [92].

Overall, class 2 CRISPR/Cas-based assays seem to have a perspective use as POCT for the detection of infectious diseases, but they still have indispensable requirements for careful experimental design, method validation, and precise data analysis.

## 5. Discussion

The specificity and sensitivity of nucleic acid testing are based on complementary base pairing, but the specificity and sensitivity of Ag-Ab serum reactions are dependent on Ag/Ab conformation and their binding affinity. Nucleic acid testing is generally regarded as more precise, specific, and sensitive than Ag-Ab serum reactions, but it is relatively, not absolutely, better than Ag-Ab serum reactions. Nucleic acid testing is sometimes even an Ag-Ab-based test for specificity and sensitivity depending upon the infectious agent. Most of the diagnostic tests described only involve relatively small sample sizes. This may strongly affect the validity of the statistical values for sensitivity, specificity, accuracy, predictive results of positive and negative tests, etc. RNA amplification techniques, including qPCR/qRT-PCR, are considered the gold standard among other diagnostic methods for COVID-19 [93,94]. Nevertheless, the gold standard for the detection of infectious diseases may be a validated and accepted test currently available for only a certain infectious agent. Additionally, specificity and sensitivity may be variable for the detection of different microorganisms using different detection methods. In some cases, qPCR/qRT-PCR is not significantly better than Ag-Ab serum reaction for the detection of infectious diseases. For example, the sensitivity of the RT-RPA with CRISPR/Cas12a assay is only 85.07% for IAV and 94.87% for IBV detection; the specificity for IAV and IBV detections was both about 96% [70]. A sensitivity of 85% means there are 15% false negatives, and a specificity of 96% indicates there are 4% false positives. If the actual disease incidence is less than 4%, then the testing may be considered not better than simply guessing significantly. Despite this, the class 2 CRISPR/Cas-based assay is rapid, portable, and easy to operate and can provide results visible by the naked eye for POCT compared with qPCR (or qRT-PCR) which is not suitable for POCT because of its time consumption, inconvenience, and requirements for special devices and professional operation.

## 6. Conclusions

The Ag-Ab serum reaction assay on a strip is quick, specific, convenient, and portable, so it can be used for POCT; however, the result is not reliable enough because its sensitivity and accuracy may be insufficient. Nucleic acid testing, including qPCR/qRT-PCR, is time-consuming, troublesome, and not portable, so it is not suitable for POCT. This is why the class 2 CRISPR/Cas-based assay is a better method for POCT compared with the Ag-Ab serum reaction assay on a strip and qPCR/qRT-PCR. Class 2 CRISPR/Cas-based nucleic acids detection can be further developed by combining engineering, materials, and other disciplines. Innovative methods have the potential to be used to identify microorganisms for the vast majority of infectious diseases caused by bacteria, viruses, fungi, and parasites. In the future, we believe that class 2 CRISPR/Cas-based assays can not only achieve amplification-free detection with high sensitivity and specificity but also have the advantages of being portable and low-cost. It may be able to satisfy the requirements of specificity, sensitivity, speed, cost, and simplicity simultaneously for POCT. 

## Figures and Tables

**Table 1 diagnostics-13-02255-t001:** Comparison of traditional methods used to detect microorganisms.

Detection Method	Nucleic Acid Testing		Antigen–Antibody (Ag-Ab) Serum Reaction	
	DNA or RNA Hybridization	Real-Time qPCR	Antigen Testing	Antibody Testing
Principle	Southern blotting: Detect microbial DNA Northern blotting: Detect microbial RNA	Detect microbial DNA or RNA	Monoclonal antibodies are used to detect microbial antigens.	Specific antigens are used to detect antibodies induced by microorganisms
Required lime	12–18 h	3–4 h	20–30 min	20–30 min
Advantages	1. Higher accuracy rate and more sensitive than Ag-Ab reactions 2. Qualitative and semiquantitative	1. Higher accuracy rate and more sensitive than Ag-Ab reactions 2. Qualitative and relatively and absolutely quantitative	1. Rapid detection of microorganisms at earlier time 2. Convenient 3. Qualitative and semiquantitative	1. Detection of infected persons 2. Detection of a prior infection and potentially to distinguish between vaccinated and naturally infected individuals. 3. Convenient 4. Qualitative and semiquantitative
Disadvantages	1. Time-consuming 2. Bulky and precise devices are needed 3. Laboratory and professional operation are required	1. Time-consuming 2. Bulky and precise devices are needed 3. Laboratory and professional operation are required	Lower accuracy rate and less sensitive than nucleic acid testing	1. Lower accuracy rate and less sensitive than nucleic acid testing 2. Detection at later time
Reference	[1,2,3,4]	[5,6,7]	[8,9,10,11]	[8,9,10,11]

## Data Availability

Not applicable.

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
