# Peer review of "Point-of-Care Testing for Infectious Diseases Based on Class 2 CRISPR/Cas Technology"

_diagnostics, 2023, doi:10.3390/diagnostics13132255_

Round 1

Reviewer 1 Report (Previous Reviewer 3)

In this revision of the manuscript the authors have made improvements that help to put some of the information (mainly descriptions of specific infectious diseases) into a more meaningful context relative to use of diagnostics. Despite these improvements some significant problems remain.

1. Confusion seems to remain about the difference between listing facts about a disease and discussing the relative merits of the class 2 Cas-based diagnostics versus other tests.  Each disease presents unique problems for diagnosis.  It would be helpful to discuss what these problems are and why the class 2 Cas-based diagnostics might work better- or not- in a given situation.  As it is, each disease is summarized with a statement that other types of tests have problems and class 2 Cas-based diagnostics could be a POCT.  What is lacking is any meaningful input as to WHY it may be superior and when it may not.  Ideally, a graduate student reading this should be able to see parallels with whatever system they’re studying and be able to discern why one of the various class 2 Cas-based diagnostics might be the one to put their efforts into developing, or not.  This review still does not provide the information or intellectual nuance to aid in such decisions; i.e., the “why” aspect remains lacking.

2. Despite all the positives that class 2 Cas-based tests might provide they, too, are not perfect.  Shortcomings and limitations are not well discussed.

3. lines 385-386. This statement about sensitivity is only true if multiple samplings are considered/ recommended as routine for detection of HCV.  If single samplings are performed the sensitivity is 93%. This issue was pointed out previously.

4. lines 481-489 make no sense.  Please rewrite.  I am also unclear as to why the authors are reluctant to invoke the rigorous definitions of sensitivity, specificity, predictive values, etc. rather than referring the reader to “..books dealing with medical statistics..”. It should be noted that most of the diagnostic tests described involved relatively small sample sizes. This strongly affects the validity of the above statistical values, yet no mention is made of confidence intervals.  The same holds for the discussion of gold standards.  After all, if the class 2 Cas-based tests are not demonstrably better than the current gold standard why would one switch?  These subtleties need to be incorporated to be convincing.

5. references 13 and 14 are duplicates.

Overall, English usage is acceptable.  There remain numerous instances of mismatch between subject and verb form.  While correcting this would aid readability this is the least troublesome of English language problems and not an impediment to publication.

Author Response

In this revision of the manuscript the authors have made improvements that help to put some of the information (mainly descriptions of specific infectious diseases) into a more meaningful context relative to use of diagnostics. Despite these improvements some significant problems remain.

  1. Confusion seems to remain about the difference between listing facts about a disease and discussing the relative merits of the class 2 Cas-based diagnostics versus other tests.  Each disease presents unique problems for diagnosis.  It would be helpful to discuss what these problems are and why the class 2 Cas-based diagnostics might work better- or not- in a given situation.  As it is, each disease is summarized with a statement that other types of tests have problems and class 2 Cas-based diagnostics could be a POCT.  What is lacking is any meaningful input as to WHY it may be superior and when it may not.  Ideally, a graduate student reading this should be able to see parallels with whatever system they’re studying and be able to discern why one of the various class 2 Cas-based diagnostics might be the one to put their efforts into developing, or not.  This review still does not provide the information or intellectual nuance to aid in such decisions; i.e., the “why” aspect remains lacking.

Ans: The most effective method to prevent the spread of diseases and guide proper treatment is to provide a quick and on-site diagnostic test which is accurate, sensitive, convenient, cheap and portable, such as the point-of-care testing (POCT). POCT can provide fast and feasible diagnostic results near patients and thereby acts as a personal exploring detector for infectious diseases. The Ag-Ab serum reaction assay on a strip is quick, specific, convenient and portable so that it can be used for POCT; however, the result is not reliable enough because its sensitivity and accuracy may be insufficient. The nucleic acid testing including qPCR/qRT-PCR is time-consuming, troublesome, and not portable, so it is not suitable for POCT. This is why the class 2 CRISPR/Cas-based assay is a better method for POCT, compared with the Ag-Ab serum reaction assay on a strip and qPCR/qRT-PCR. (P.2 line 65-P.3 line 72; P.14 line 518-530)

  1. Despite all the positives that class 2 Cas-based tests might provide they, too, are not perfect. Shortcomings and limitations are not well discussed.

Ans: We have revised the statement to well discuss the limitations of class 2 CRISPR/Cas-based assay. (P.12 line 432-P.13 line 489)

  1. lines 385-386. This statement about sensitivity is only true if multiple samplings are considered/ recommended as routine for detection of HCV.  If single samplings are performed the sensitivity is 93%. This issue was pointed out previously.

Ans: We have revised the statement into “This means the sensitivity of the first round and the second round is respectively 93% and 96% in both readouts of the RT-LAMP-coupled CRISPR–Cas12 assay.” (P.10 line 376-378)

  1. lines 481-489 make no sense.  Please rewrite.  I am also unclear as to why the authors are reluctant to invoke the rigorous definitions of sensitivity, specificity, predictive values, etc. rather than referring the reader to “..books dealing with medical statistics..”. It should be noted that most of the diagnostic tests described involved relatively small sample sizes. This strongly affects the validity of the above statistical values, yet no mention is made of confidence intervals.  The same holds for the discussion of gold standards.  After all, if the class 2 Cas-based tests are not demonstrably better than the current gold standard why would one switch?  These subtleties need to be incorporated to be convincing.

Ans: We have rewriten the statement to demonstrate that the class 2 CRISPR/Cas-based tests are better assay for POCT than qPCR/qRT-PCR which is considered the current gold standard for detection. (P.13 line 498-P.14 line 520)

  1. references 13 and 14 are duplicates.

Ans: We have replaced the original reference Reference 14 with a new reference.

Comments on the Quality of English Language

Overall, English usage is acceptable.  There remain numerous instances of mismatch between subject and verb form.  While correcting this would aid readability this is the least troublesome of English language problems and not an impediment to publication.

Ans: We have tried our best to check the grammar of this manuscript and revise the grammatical error for publication.

Reviewer 2 Report (Previous Reviewer 1)

Yes  dear  I  saw  revised  article  ,, now  it is  ok 

the  author  have done  all   corrections  

now  it is suitable  for  this  journal 

Author Response

Yes  dear  I  saw  revised  article ,, now  it is  ok. The  author  have done  all   corrections, now  it is suitable  for  this  journal 

Ans: We greatly appreciate the revierer’s comment and will try our best to revise the manuscript for publication.

This manuscript is a resubmission of an earlier submission. The following is a list of the peer review reports and author responses from that submission.

Round 1

Reviewer 1 Report

Dear   Authors . it is  good  scientific  study  ,,  but  to  be  acceptable , it  needs  Minor  corrections :

1- There  is  self citation  in  reference  No.  (25).

2-  Table (2)  needs  more  explanation  of  Comparison  to  improve  current  results

3-  Also    Table (1)  needs  clarification 

4-  ((3.1. Acute Respiratory Syndrome Coronavirus 2 (SARS-CoV-2)))needs   more  clarification  in  explanation  ,, it  is  not  clear  for  reader 

5- Also  (( 3.6. Hepatitis C (HCV) ))  is  not  clear  for  reader ,,, it  needs  more  clarification 

6-  Replace   reference  No. 52      by  other 

7- I accepted  paper  after   Minor  Corrections  

Author Response

Reviewer 1:

Dear   Authors . it is  good  scientific  study , but  to  be  acceptable , it  needs  Minor  corrections :

  1. There  is  self citation  in  reference  No. (25).

Ans: We have revised the paragraph and replaced the original Reference no.25~27 with new ones (P.4, Line 157~P.5, Line 164).

  1. Table (2)  needs  more  explanation  of  Comparison  to  improve  current  results.

Ans: We have revised Table 2 (P.5, Line 175~176).

  1. Also    Table (1)  needs  clarification 

Ans: We have revised Table 1 (P.2, Line55~56).

  1. ((3.1. Acute Respiratory Syndrome Coronavirus 2 (SARS-CoV-2)))needs   more  clarification  in  explanation, it  is  not  clear  for  reader.

Ans: We have provided some statement about SARS-CoV-2 to clarify it (P.6, Line184~188).

  1. Also  (( 3.6. Hepatitis C (HCV) ))  is  not  clear  for  reader ,,, it  needs  more  clarification.

Ans: We have provided some statement about HCV to clarify it (P.9, Line 319~335).

  1. Replace   reference  No. 52 by  other 

Ans: We have replaced the old Reference 52 with a new one (P.10, Line 369~373; P.14, Line 578~584, Reference 54, 55).

  1. I accepted  paper  after   Minor  Corrections

Ans: We thanks for the reviewer’s comment and try our best to improve this manuscript.

Reviewer 2 Report

The authors propose a review of point-of-care tests based on CRISPR/Cas technology for the diagnosis of infectious diseases. The manuscript is well written, easy to read, with a coherent thread, and very timely and updated information. 

However, below are some points that are not clear enough.

>Line 51 and the disadvantage section of Table 1. It is mentioned that the sensitivity and accuracy of tests based on antigens and antibodies are not reliable and have low accuracy. First, it is difficult to determine the sensitivity and accuracy of antigen-antibody tests for all infectious diseases. For some infectious diseases, the sensitivity and specificity are high 98 to 100%. So this categorical phrase is not correct without supporting it with the respective references. 

> Lines 331-332, This phrase is meaningless or incomplete

> Lines 338-339, Again it is stated categorically that all tests based on CRISPR/Cas are more precise, sensitive, and specific than all tests based on Antigens-Antibodies. This contradicts the results presented above where CRISPR/Cas-based tests have a low sensitivity of 85% (line 237). Nor do I agree that the speed of testing is "almost the same". It is still far from real.

>Lines 348-350, This phrase is confusing and does not reveal what the cited references expose

Author Response

Reviewer 2:

The authors propose a review of point-of-care tests based on CRISPR/Cas technology for the diagnosis of infectious diseases. The manuscript is well written, easy to read, with a coherent thread, and very timely and updated information. 

However, below are some points that are not clear enough.

>Line 51 and the disadvantage section of Table 1. It is mentioned that the sensitivity and accuracy of tests based on antigens and antibodies are not reliable and have low accuracy. First, it is difficult to determine the sensitivity and accuracy of antigen-antibody tests for all infectious diseases. For some infectious diseases, the sensitivity and specificity are high 98 to 100%. So this categorical phrase is not correct without supporting it with the respective references. 

Ans: We have revised the statement about sensitivity and specificity in Table 1 (P.2, Line55~56).

> Lines 331-332, This phrase is meaningless or incomplete.

Ans: We have revised this phrae into “Currently, these technologies are not only under basic research or clinical trial, but also some of them are advanced to clinical application.” (P.10, Line 366~368)

> Lines 338-339, Again it is stated categorically that all tests based on CRISPR/Cas are more precise, sensitive, and specific than all tests based on Antigens-Antibodies. This contradicts the results presented above where CRISPR/Cas-based tests have a low sensitivity of 85% (line 237). Nor do I agree that the speed of testing is "almost the same". It is still far from real.

Ans: We have revised the statement into “The diagnostic rapidity of CRISPR/Cas-based assay can be the same as the Ag-Ab serum reaction in most cases, but its accuracy, specifcity and sensitivity is generally better than the Ag-Ab serum reaction in detecting microorganisms.” (P.10, Line 374~377)

>Lines 348-350, This phrase is confusing and does not reveal what the cited references expose

Ans: We have revised the statement into “Fortunately, a novel CRISPR/Cas12-base assay requiring only a miniature heater has been established and can replace fluorescence microscopy with mobile phone-based fluorescence for quantitative detection.” (P.10, Line 385~388)

Reviewer 3 Report

POC diagnostics for infectious diseases are a valuable tool in the diagnosis and treatment of patients, and in certain epidemiologic studies. Yet, a paucity of such tools exists. The authors provide in this manuscript a review of recently published POC diagnostics based upon CRISPR/Cas systems. The title is appropriate, the Abstract accurately reflects content, and overall the usage of English is acceptable (but would benefit from the use of spell-check). Improvement of the below shortcomings prior to publication would greatly strengthen the value of this manuscript.

Major concerns

1. As currently written, this manuscript would not be very helpful to someone interested in developing a POC diagnostic based upon a CRISPR/Cas system. The Introduction gives a generally adequate discussion of the weaknesses of other classes of POC tests and a description of the various CRISPR/Cas systems. However, the authors then provide mainly a laundry list of publications from the past couple of years, in which they repeatedly briefly describe a “test”, provide its sensitivity and specificity, and generally state that it is a good test. There is no discernible rigor applied in analyzing the relative strengths/ weaknesses/ advantages/ disadvantages of the various test formats, nor any real discussion of these topics. A focused, rigorous discussion of these topics could help to guide someone new to the field (where reviews are most helpful) in deciding what sort of format might work best for their system based upon the identified characteristics and trade-offs of the various systems.  This was not provided.

2. The authors compare the performance of most of the described tests against a “gold standard” qPCR or RT-qPCR test, yet never identify the qPCR or RT-qPCR as such, or describe the necessity, function, or relative strengths/ weaknesses of gold standards.  If the gold standard is not actually very good, the fact the CRISPR/Cas system performs similarly is not impressive.  This concept should be introduced and used rigorously.

3. The authors use the terms of sensitivity, specificity, and positive and negative predictive values without defining them or considering the effects of disease incidence has on the value of each.  This is very important. For example, in line 237 they give a sensitivity of 85% for Influenza A, then in line 239 unjustifiably call the test “an effective platform”.  A sensitivity of 85% means there are 15% false negatives, and with a specificity of 96% there are 4% false positives.  If actual disease incidence is less than 4% then the test is effectively worse than simply guessing.  It would be helpful if the authors were to treat such shortcomings honestly, along with discussing why some formats might function better than others, and the circumstances under which it may be acceptable to use a test with such characteristics.

Minor points

4. line 23. delete “..is required”.

5. line 29. Consider replacing “..potential for becoming perfect..” with “..development of..”.

6. line 71. It is entirely unclear what is meant by, “..special and regular sequence..”.  Please reword for clarity.

7. lines 97-102.  The authors seem to confuse the in vivo activity of Cas9 involving separate crRNA and guide RNA (gRNA) molecules with the in vitro experimental use of Cas9 with a single guide RNA (sgRNA).  That change from a tripartite to a bipartite system was crucial to making Cas9 practical to use for genome editing and part of the reason Doudna and Charpentier received the Nobel prize.  It also makes the potential use for a POC diagnostic feasible.

8. line 226. The authors describe a test having a LOD of 0.152 pM over a range of from 10 pM-100 nM.  This must be an error.  Please correct.

9. line 246. Several of the described tests had sensitivities of 1 copy ml-1 or even less.  Here, the authors describe 500 copies ml-1 as “high sensitivity”.  For some microbes, that 2 orders of magnitude difference could make a profound difference in disease outcome.  Some justification for what is called high and low sensitivity should be provided.

Overall, the quality of English is acceptable even though occasionally a little awkward.  However, spell-check needs to be applied as there were several misspellings noticed.

Author Response

Reviewer 3:

POC diagnostics for infectious diseases are a valuable tool in the diagnosis and treatment of patients, and in certain epidemiologic studies. Yet, a paucity of such tools exists. The authors provide in this manuscript a review of recently published POC diagnostics based upon CRISPR/Cas systems. The title is appropriate, the Abstract accurately reflects content, and overall the usage of English is acceptable (but would benefit from the use of spell-check). Improvement of the below shortcomings prior to publication would greatly strengthen the value of this manuscript.

Major concerns

  1. As currently written, this manuscript would not be very helpful to someone interested in developing a POC diagnostic based upon a CRISPR/Cas system. The Introduction gives a generally adequate discussion of the weaknesses of other classes of POC tests and a description of the various CRISPR/Cas systems. However, the authors then provide mainly a laundry list of publications from the past couple of years, in which they repeatedly briefly describe a “test”, provide its sensitivity and specificity, and generally state that it is a good test. There is no discernible rigor applied in analyzing the relative strengths/ weaknesses/ advantages/ disadvantages of the various test formats, nor any real discussion of these topics. A focused, rigorous discussion of these topics could help to guide someone new to the field (where reviews are most helpful) in deciding what sort of format might work best for their system based upon the identified characteristics and trade-offs of the various systems.  This was not provided.

Ans: We have added a section of Discussion to analyze the relative strengths/ weaknesses/ advantages/ disadvantages of the various test formats, along with discussing why some formats might function better than others. (P.11, Line 400~418)

  1. The authors compare the performance of most of the described tests against a “gold standard” qPCR or RT-qPCR test, yet never identify the qPCR or RT-qPCR as such, or describe the necessity, function, or relative strengths/ weaknesses of gold standards.  If the gold standard is not actually very good, the fact the CRISPR/Cas system performs similarly is not impressive.  This concept should be introduced and used rigorously.

Ans: We have added a paragraph to describe the necessity, function, or relative strengths/ weaknesses of gold standards. (P.2, Line 57~68) 

  1. The authors use the terms of sensitivity, specificity, and positive and negative predictive values without defining them or considering the effects of disease incidence has on the value of each.  This is very important. For example, in line 237 they give a sensitivity of 85% for Influenza A, then in line 239 unjustifiably call the test “an effective platform”.  A sensitivity of 85% means there are 15% false negatives, and with a specificity of 96% there are 4% false positives. If actual disease incidence is less than 4% then the test is effectively worse than simply guessing.  It would be helpful if the authors were to treat such shortcomings honestly, along with discussing why some formats might function better than others, and the circumstances under which it may be acceptable to use a test with such characteristics.

Ans: We have remove the word “effective” in line 239. Additionally, we have added a section of Discussion to analyze the relative strengths/ weaknesses/ advantages/ disadvantages of the various test formats, along with discussing why some formats might function better than others. (P.11, Line 400~418)

Minor points

  1. line 23. delete “..is required”.

Ans: We have rephrased the sentence as “A quick and on-site diagnostic test (e.g., point-of-care testing) is an efficient method for the prevention of spreading and treatment of infectious diseases.” (P.1, Line 22~23)

  1. line 29. Consider replacing “..potential for becoming perfect..” with “..development of..”.

Ans: We have rephrased the sentence as “The powerful capacity of these methods will facilitate the development of diagnostic tools for point-of-care testing, though they still have some limitations.” (P.1, Line 30~31)

  1. line 71. It is entirely unclear what is meant by, “..special and regular sequence..”.  Please reword for clarity.

Ans: We have removed the words “special and” and rephrased as “regular sequence”. (P.3, Line 87)

  1. lines 97-102.  The authors seem to confuse the in vivo activity of Cas9 involving separate crRNA and guide RNA (gRNA) molecules with the in vitro experimental use of Cas9 with a single guide RNA (sgRNA).  That change from a tripartite to a bipartite system was crucial to making Cas9 practical to use for genome editing and part of the reason Doudna and Charpentier received the Nobel prize.  It also makes the potential use for a POC diagnostic feasible.

Ans: We revised the description for this paragraph and Table 2 including adding references. (P.5, Line 175~176)

  1. line 226. The authors describe a test having a LOD of 0.152 pM over a range of from 10 pM-100 nM.  This must be an error.  Please correct.

Ans: We have corrected it as “The colorimetric biosensor showed linear relationship over the range from 10 pM to 100 nM. The method has excellent specifcity and sensitivity with a LOD of 0.152 pM.” (P.7,  Line 253~256)

  1. line 246. Several of the described tests had sensitivities of 1 copy/ml or even less.  Here, the authors describe 500 copies/ml as “high sensitivity”.  For some microbes, that 2 orders of magnitude difference could make a profound difference in disease outcome.  Some justification for what is called high and low sensitivity should be provided.

Ans: We have removed the word “high” in “high sensitivity ” and corrected it into “ sensitivity”. (P.5, Line 275)

Round 2

Reviewer 3 Report

This is a revised manuscript in which the authors have attempted to respond to criticisms of the original, for which they are thanked. However, significant deficiencies remain that continue to reduce the overall value and contribution of this manuscript.

1. Table 1. In contrasting nucleic acid and antibody-based testing there are significant advantages to antibody-based tests that the authors fail to mention, namely the abilities to detect a prior infection and potentially to distinguish between vaccinated and naturally-infected individuals.  These can also complicate diagnosis. Although this review is focused on CRISPR/Cas-based diagnostics the rationales behind these differences, and when nucleic acid-based tests improve on the situation, should be discussed for an unbiased comparison.

2. lines 57-68. This is a nice discussion of the use of qRT-PCR as a gold standard. However, qRT-PCR is not the gold standard for all disease agents. Rather, the gold standard is the best validated and accepted test currently available for that agent. Depending upon the agent that is sometimes even an antibody-based test. While the authors are commended for adding this discussion it is misleading and remains quite incomplete.

3. line 255. It is unclear how the LOD was quantified to 3 significant digits when the standards are given to 1 significant digit.

4. lines 324-335. All the newly added detail provides only marginal information, and does not make it apparent that to achieve 96-97% sensitivity required repeated samplings, a point the authors fail to bring up. This would be a real problem for a POC diagnostic that is not appreciated in this discussion. It is not possible to state specificity from the information provided without adding the number of false positives.

5. lines 400-418. The authors have failed to provide a meaningful discussion of the relative merits of nucleic acid-based tests, why some might work better or worse under certain circumstances, etc. In other words, useful information to guide researchers in making decisions what tests to invest time and energy into developing, and why, is lacking. The authors should be aware that there are, in fact, rigorous definitions of sensitivity, specificity, accuracy, predictive value of positive and negative tests, etc. that may be referred to in any book dealing with medical statistics, along with discussions of gold standards of various sorts. For a straightforward, easily generalizable treatment of the topic, Courtney and Cornell. 1990. JAVMA 197: 724 is a good place to start.

Overall the usage of English is appropriate and easily understandable.  However, the use of spellcheck (and grammarcheck if available) is still advised.

Author Response

This is a revised manuscript in which the authors have attempted to respond to criticisms of the original, for which they are thanked. However, significant deficiencies remain that continue to reduce the overall value and contribution of this manuscript.

  1. Table 1. In contrasting nucleic acid and antibody-based testing there are significant advantages to antibody-based tests that the authors fail to mention, namely the abilities to detect a prior infection and potentially to distinguish between vaccinated and naturally-infected individuals.  These can also complicate diagnosis. Although this review is focused on CRISPR/Cas-based diagnostics the rationales behind these differences, and when nucleic acid-based tests improve on the situation, should be discussed for an unbiased comparison.

Ans: We think “infection tracking” is the same meaning “as to detect a prior infection and potentially to distinguish between vaccinated and naturally-infected individuals.” Therefore, we have replaced infection tracking with “Detection of a prior infection and potentially to distinguish between vaccinated and naturally-infected individuals.” (P2, Table 1)

  1. lines 57-68. This is a nice discussion of the use of qRT-PCR as a gold standard. However, qRT-PCR is not the gold standard for all disease agents. Rather, the gold standard is the best validated and accepted test currently available for that agent. Depending upon the agent that is sometimes even an antibody-based test. While the authors are commended for adding this discussion it is misleading and remains quite incomplete.

Ans: We have replaced “the gold standard” with “a better method” (P.2, line 64). Additionally, we added “The nucleic acid testing is sometimes even an antibody-based test for specificity and sensitivity, depending upon the infectious agent.” In the discussion section. (P.11, line 401~403)

  1. line 255. It is unclear how the LOD was quantified to 3 significant digits when the standards are given to 1 significant digit.

Ans: We cited these data from reference 45. The colorimetric biosensor exhibited a linear relationship from 10 pM to 100 nM. The detection limit was 0.152 pM. It is reasonable that the linear range (1 significant digit) has different significant digits from LOD (3 significant digit), because they stand for different type of data.

  1. lines 324-335. All the newly added detail provides only marginal information, and does not make it apparent that to achieve 96-97% sensitivity required repeated samplings, a point the authors fail to bring up. This would be a real problem for a POC diagnostic that is not appreciated in this discussion. It is not possible to state specificity from the information provided without adding the number of false positives.

Ans: We have revised the discussion section. (P.11, Line 323~326)

  1. lines 400-418. The authors have failed to provide a meaningful discussion of the relative merits of nucleic acid-based tests, why some might work better or worse under certain circumstances, etc. In other words, useful information to guide researchers in making decisions what tests to invest time and energy into developing, and why, is lacking. The authors should be aware that there are, in fact, rigorous definitions of sensitivity, specificity, accuracy, predictive value of positive and negative tests, etc. that may be referred to in any book dealing with medical statistics, along with discussions of gold standards of various sorts. For a straightforward, easily generalizable treatment of the topic, Courtney and Cornell. 1990. JAVMA 197: 724 is a good place to start.

Ans: We have revised the discussion section. (P.11, Line 401~408)